# Carbohydrate Sources Influence the Microbiota and Flavour Profile of a Lupine-Based Moromi Fermentation

**DOI:** 10.3390/foods12010197

**Published:** 2023-01-02

**Authors:** Rebekka H. Lülf, Karl Selg-Mann, Thomas Hoffmann, Tingting Zheng, Melanie Schirmer, Matthias A. Ehrmann

**Affiliations:** 1Chair of Microbiology, Technical University of Munich (TUM), 85354 Freising, Germany; 2Abteilung Qualität und Forschung, Purvegan GmbH, 67305 Ramsen, Germany; 3Associate Professorship of Biotechnology of Natural Products, Technical University of Munich (TUM), 85354 Freising, Germany; 4ZIEL—Institute for Food & Health, Technical University of Munich (TUM), 85354 Freising, Germany

**Keywords:** lupine moromi, sensory analysis, GC-MS, *Zygosaccharomyces rouxii*, *Debaryomyces hansenii*

## Abstract

Lupine-based seasoning sauce is produced similarly to soy sauces and therefore generates a comparable microbiota and aroma profile. While the koji state is dominated by *Aspergillus oryzae*, the microbiome of the moromi differs to soy moromi, especially in yeast composition due to the absence of *Zygosaccharomyces rouxii* and *Debaryomyces hansenii* as the dominant yeast. In this study, we monitored the addition of a carbohydrate source on the microbiome and aroma profile of the resulting sauce. Compared to previous studies, the usage of a yeast starter culture resulted in a sparsely diverse microbiota that was dominated by *D. hansenii* and *T. halophilus*. This led to a pH below 5 even after four months of incubation and most of the measured aroma compounds were pyrazines and acids. The addition of wheat and buckwheat resulted in a temporary change in the yeast consortium with the appearance of *Z. rouxii* and additional bacterial genera. The aroma profile differs in the presence of pyrazines and esters. Since no significant differences in the taste and odour of wheat-added and buckwheat-added sauce was sensed, both substrates influence the lupine sauce in a similar way.

## 1. Introduction

Fermented seasoning sauces such as soy and fish sauces are very popular to enhance the saltiness and umami taste of food all over the world [1]. Recently, plant-based alternatives are becoming more and more attractive for the industry. Compared to soy beans, lupine seeds contain a similar protein content but higher amounts of carbohydrates, which may make them a suitable alternative without the requirement of wheat addition [2]. Parallel to soy sauces, lupine-based seasoning sauce is produced in a two-step fermentation process.

For the initial step, the so-called koji, lupine seeds are roasted and crushed, and inoculated with *Aspergillus oryzae* for two days. With the addition of brine at 13.5% sodium chloride, the growth of the mould and potentially pathogenic microorganisms is inhibited, and this second fermentation step is called moromi (mash). Additionally, the microbiota can be modified using starter cultures as well as altered NaCl levels [3,4]. This is incubated for six months and lactic acid bacteria such as *Weissella paramesenteroides*, *Tetragenococcus halophilus*, *Chromohalobacter* species, and *Staphylococcus equorum*, and yeasts such as *Candida guilliermondii* and *Debaryomyces hansenii* are then dominating the moromi [2].

Previously, *D. hansenii* was isolated from the early phase of fermentation of *Ganjang*, a Korean soy sauce from a naturally fermented soybean block called *Meju* [5]. It also temporarily dominates the moromi-like *baceman* state of the Indonesian sauce *kecap manis* which is based on black soybeans [6]. Furthermore, *D. hansenii* is associated with other foods such as cheese, raw sausage, wine, beer, and fruit [7]. They produce various flavour components, such as branched-chain aldehydes and alcohols [8].

In Japanese wheat-containing soy moromi, the most dominant yeast is *Zygosaccharomyces rouxii* [9]. It grows at a pH below 5 and uses glucose to produce alcohols such as ethanol and fusel alcohols [10,11]. Moreover, it is an important producer of flavour compounds such as 4-hydroxyfuranones and therefore, its absence in lupine moromi is further investigated in this study. Since the absence of *Z. rouxii* was often mentioned for moromi based solely on soy beans [9], the addition of wheat as the standard carbohydrate source in soy moromi, and buckwheat as a gluten-free alternative were tested in this study. As we wanted to compare the persistence of *D. hansenii* and *Z. rouxii*, a mixed starter culture was inoculated in the moromi.

## 2. Materials and Methods

### 2.1. Koji and Moromi Fermentation

For the koji, lupine seeds (*Lupinus angustifolius*), wheat grains, and buckwheat were roasted separately at 150 °C for 30 min. Each of these was mixed up with 70% soaked lupine seeds and merged with spores of *Aspergillus oryzae*. Incubation was executed at 25–35 °C for two days. Each moromi batch consisted of 40% koji, either with roasted lupine seeds, wheat, or buckwheat, and brine with 13.5% (*w*/*v*) NaCl. As starting cultures, 10^5^ CFU/mL of *D. hansenii* strain TMW 3.1188 and *Z. rouxii* strain DSM 2531 were added. Each fermentation was prepared in duplicate and incubated at 25 °C at a humidity of 80%. A weekly mixing was executed for ventilation and the volume level was kept stable with deionised water.

### 2.2. pH Measurement, Cell Counts and Chromatography

The pH measurements, cell counts, and gas chromatography–mass spectrometry (GC-MS) were completed as in previous works [2]. We used malt extract peptone agar (ME, malt extract 20 g/L, soy peptone 2 g/L, NaCl 25 g/L, agar 15 g/L) at pH 5.6 and incubated under oxic conditions at 30 °C. Several dilutions of the samples were plated so that a standard deviation of cell counts could be calculated from quadruplicates. All colonies larger than 0.5 mm were recorded.

Furanones were detected via HPLC-MS from moromi supernatants in duplicates. For the extraction, an 8 g sample was mixed up with 25 mL H_2_O and 10 µL phenol (10 mg/mL, internal standard) and centrifuged (5000× *g*, 10 min). The solid residue was re-extracted by 25 mL H_2_O, centrifuged, and the combined supernatants were subjected to solid phase extraction on Amberlite XAD-2 polymeric adsorbent (20–60 mesh; Merck, Darmstadt, Germany). The column was rinsed with 50 mL H_2_O and afterwards compounds were eluted with 100 mL diethyl ether. The diethyl ether extract was dried with sodium sulphate, concentrated by using a rotary evaporator, constricted to dryness with a stream of nitrogen and transferred into 100 µL of water. The aqueous phase was analysed by HPLC.

The HPLC system used was an Agilent 1200 HPLC system composed of two Isocratic Pumps (G1310A), a Micro Well-plate Autosampler (G1377A), a Column Thermostat (G1316A), a Diode Array Detector SL (G1315C), and an Agilent 6320 Ion Trap. The column was a LUNA C18(2) (150 mm × 2.0 mm, particle size 5 µm; Phenomenex; Aschaffenburg, Germany) held at 28 °C. The binary gradient system consisted of solvent A, water with 0.1% formic acid and solvent B, 100% methanol with 0.1% formic acid. The gradient program was as follows: 0–10 min, 100% A to 90% A/10% B; 10–50 min, 90% A/10% B to 70% A/30% B; 50–65 min, 70% A/30% B to 20% A/80% B; 65–70 min, 100% B; 70–80 min, 100% A. The flow rate was 0.2 mL/min and the injection volume was 5 µL.

Compounds were identified by their retention times and mass spectra in comparison with data determined for authentic reference materials. Quantification was achieved by means of calibration curves and DAD (UV traces 272 nm for Phenol, 290 nm for Furaneol and 284 nm for Norfuraneol).

### 2.3. Identification of Organisms

Identification of yeasts at species level were based on low molecular sub-proteome profiling using matrix-assisted laser desorption–ionisation time-of-flight mass spectrometry (MALDI-TOF-MS). A total of 96 colonies per sample were streaked at least once and measured with Microflex LT Spectrometer (Bruker Daltonics, Bremen, Germany). Identification was achieved via comparison of detected spectra with the Bruker Biotyper database (Bruker Daltonics) and an in-house database validated with 18S rRNA gene sequenced strains and published type strains. Moulds were identified by microscopic analysis of hyphae and spores by Prof. Dr. Ludwig Niessen.

Bacterial samples were analysed along with yeasts by MALDI-TOF-MS and via 16S ribosomal RNA gene amplicon sequencing by Eurofins Genomics Europe Sequencing GmbH (Konstanz, Germany). Therefore, the V4 region was amplified from purified DNA using primers 519F (5′ CAG CMG CCG CGG TAA TWC) and 785R (5′ GAC TAC HVG GGT ATC TAA TCC) [12]. Sample preparation and analysis procedure were completed according to the method described previously [2].

### 2.4. Sensory Analysis

The different lupine sauce variants were tested by a panel for the perceptible flavour components and their intensity.

The panel members were selected from a group of 15 people who had to describe the aroma impression of eight different smelling sticks with defined aroma substances and classify their intensity using a scale. Only people who had sufficient sensitivity to the majority of these aromas were then included in the panel. The final panel then consisted of 10 people (3 female, 7 male). A total of seven of these people completed training in the food sector or were working in the sector for several years.

A list of typical soy sauce flavours was used as the template for a questionnaire [13]. The list was supplemented with a few more terms based on the results of GC analyses; similar and synonymous terms were removed as far as possible. A questionnaire with 35 aroma and flavour terms (taste and smell) remained. The participants received 1 mL samples of each variant in a 25 mL beaker to smell and taste. If a perception could be assigned to a concept, a number from 1 to 4 was assigned for the intensity (1 = very weak, hardly perceptible to 4 = very strong, dominating). Statistical tests on the pairwise comparisons of the sensory data of the biological replicates, as well as the aroma profiles of lupine sauces with additional carbohydrate sources, were performed using the Wilcoxon test.

## 3. Results

### 3.1. Major Characteristics of Lupine-Based Moromi Fermentation

The microbial cell counts and pH values changed during the fermentation and differed between the different batches (Figure 1). They all started with 1.7–2.4 × 10^7^ CFU/mL and reached 1.4–8.0 × 10^9^ CFU/mL after 16 weeks of incubation. Within the first four weeks in the fermentation with lupine seeds as sole substrate, the abundance of microorganisms increased with an average recorded growth rate of 7.4 per week for batch ML1 and of 8.2 per week for ML2. Subsequently, the stationary growth phase was reached. With additional carbohydrate sources, similar growth rates were detected within the first two weeks. However, after four weeks of incubation, cell counts were lower in these batches compared to the control group which correlates with the presence of *Z. rouxii* (Figure 2). Interestingly, they all reached cell counts of similar CFU/mL after 16 weeks.

In the same way as the cell counts, the pH values were similar at the beginning of the fermentation (approx. 6.2) and also after 16 weeks (approx. 4.7). During the first four weeks, the fermentation without wheat and buckwheat showed a fast decrease below a pH of 5. For the batches with wheat and buckwheat, these low pH values were firstly detected after eight to 12 weeks. The decrease in the pH matched with the appearance of *T. halophilus*, since it appeared in ML1 and ML2 after four weeks and was detected for the first time in batches with wheat and buckwheat after eight weeks (Figure 2).

### 3.2. Microbial Composition

For the identification of microorganisms, single colonies were analysed via MALDI-TOF-MS profiling. At the beginning of the fermentation, the plates were overgrown by *A. oryzae* which was the koji starter not yet inhibited by the salt addition. Therefore, it was barely possible to detect any other organisms.

After two weeks of moromi incubation with the lupine seeds as sole substrate, *D. hansenii* was detected (Figure 2a). In the following weeks, *T. halophilus* appeared and dominated the moromi until the end of the fermentation. Throughout the fermentation, no other organisms were identified.

In the wheat-containing batches, *Z. rouxii* was detected between week two and eight with its highest abundance (1.61 × 10^7^ CFU/mL) after 2 weeks in batch ML2 (Figure 2b). The total amount of *Z. rouxii* is similar at any point, in both batches and after 2, 4, and 8 weeks (in ML2).

*D. hansenii* showed a decrease in cell counts in some samples which led to the drop of total cell counts and to a high relative abundance of *Z. rouxii*.

The addition of buckwheat led to a similar microbiota compared to wheat-added fermentations, but growth of *D. hansenii* was less severely suppressed (Figure 2c). The highest cell count of *Z. rouxii* (2.08 × 10^7^ CFU/mL) was determined after four weeks of incubation in batch MB2. After eight weeks, this species was no longer detected in fermentations with buckwheat. Independent of the addition of a carbohydrate source, all batches were dominated by *T. halophilus* and *D. hansenii* after 16 weeks of fermentation.

The 16S rDNA amplicon sequencing revealed the presence of other bacterial genera in carbohydrate-supplemented moromi after four weeks of incubation (Figure 3). Besides *Tetragenococcus*, the most abundant organisms detected were *Enterobacter* (highest sequence similarity with *E. hormaechei*, and *E. cloacae*), *Lactococcus* (*lactis*, *taiwanensis*), *Leuconostoc* (*holzapfelii*, *falkenbergense*), *Pseudomonas* (*juntendi/putida*), *Staphylococcus* (*warneri*, *aureus*, *caprae*), and *Weissella* (*confusa*, *paramesenterioudes*). It is probable that these were not detected via MALDI-TOF-MS due to the detection limit for the cultivation method (<10^2^ CFU/mL) or due to poor growth in media used in this study. In the same way as the culture-dependent method, 16S rDNA amplicon sequence analysis revealed *Tetragenococcus* as the only bacterium present at the end of the fermentation in all batches.

### 3.3. Aromatic Compounds

The aroma profile is one of the most important parameters in the evaluation of the quality of a seasoning sauce. The effect of fermentation on the aroma compound composition was determined by GC-MS analysis. In total, 108 volatile compounds were detected and a representative selection of acids, esters, ether, pyrazines, and furan(on)es are shown in Table 1. During the fermentation, acids were formed in all batches. Esters and dimethyl ether were not detected at the beginning of the fermentation but appeared, especially in fermentations with wheat and buckwheat, during the four months of incubation.

At the beginning, a number of different pyrazines were determined. However, they were metabolised during the fermentation with additional carbohydrate sources. The 2-Phenylethanol, 3-methyl-1-butanol, and benzene acetaldehyde (not shown) were measured in all batches during the whole fermentation. At the end of the fermentation, moromi with lupine seeds as sole substrate contained primarily pyrazines and acids and the volatile compounds of moromi with additional wheat and buckwheat were mainly esters and acids.

Since 4-hydroxyfuranones were not detected via GC-MS, we analysed 4-hydroxy-2,5-dimethyl-3(2*H*)-furanone (HDMF) and 4-hydroxy-5-methylfuran-3(*2H*)-one (HMF) via HPLC in matured moromi (Table 2). The 4-Hydroxy-2-ethyl-5-methyl-3(*2H*)-furanone (HEMF) quantities were below the detection limit.

HDMF was present in all moromi batches in similar amounts. However, the concentration of HMF was increased in fermentations with wheat and buckwheat compared to sole lupine seeds.

### 3.4. Sensory Analysis

For a description of the flavour, a sensory analysis of the resulting seasoning sauces was performed. Since the sensory data of the biological replicates differ from each other (Appendix A), they will be described separately. The taste was described using the six attributes of sweet, umami, bitter, astringent, salty, and sour (Figure 4) with intensities of 0 (not perceptible) to 4 (very strong).

All products were described with similar strong saltiness (2.1–2.8) and umami taste (1.3–2.0). None of the sauces was described as more than faintly astringent or bitter. The taste attributes of these products were not significantly different.

However, the seasoning sauce without additional carbohydrate sources showed a quite intense aroma of beeswax and chocolate but alcoholic, sweaty, goat-like, or wine-like flavours were not perceptible (Figure 5). The sauces with wheat showed a fermented, cheesy and nutty aroma and no floral, mouldy or hay-like flavour. For sauces with buckwheat the most dominant flavours were cheesy and fermented. Except for alcoholic, all aroma attributes were perceived in low amounts in at least one of the replicates with buckwheat. Therefore, it appears to have a complex, hard to describe aroma profile using these attributes. Overall, the lupine sauces with additional carbohydrate sources did not show significant differences (Appendix A).

## 4. Discussion

There are several options how to influence the microbiota and flavour of a fermentation. In a lupine-based moromi, higher salt content leads to a less diverse microbiota and a delay in the appearance of several organisms [2]. Here, we could show that the usage of yeasts as starter cultures resulted in an early dominance of *D. hansenii* and *T. halophilus*. In moromi solely from lupine seeds, scarcely no other organisms were detectable. Similar effects of the inhibition of bacterial growth by yeast starters in moromi were observed by Song et al. [3] in Korean soy moromi. The absence of several species might also be the reason for the lack of pH increase that was measured at the end of the fermentation in backslopped lupine moromi [2]. In several soy moromi analyses, the pH remained constantly low; Therefore, we assume that a pH increase is not beneficial for a high-quality product [4,16,17].

The addition of wheat and buckwheat led to a temporary growth of *Z. rouxii* along with a decrease in CFUs of *D. hansenii*. We assumed that higher amounts of wheat or buckwheat would elongate the presence of *Z. rouxii* in the fermentation. The lack of specific sugars might explain the absence of *Z. rouxii* in sole lupine moromi [18]. Röling et al. [6] held the presence of galactose as the sole sugar in wheat-free *kecap* fermentation was responsible for the absence of *Z. rouxii*. Despite the higher tolerance of *D. hansenii* to several stresses [19], *Z. rouxii* appears to suppress its growth temporarily in some cases. The inhibition could not be explained by the bacterial consortium, since it was similar in all carbohydrate-added batches.

The aroma profile of wheat- and buckwheat-added moromi showed a lot of different esters and acids, whereas pure lupine moromi contained predominantly pyrazines and acids. Since *T. halophilus* dominated the microbiota in all batches we assumed that these acids and the resulting decreased pH were mainly caused by *T. halophilus* [20]. Yeasts, probably *Z. rouxii*, formed esters from alcohols and acids that are usually known to enhance the fruity character in foods [21,22]. Pyrazines may contribute to a variety of flavours such as potato, popcorn, fermented soybeans, and cocoa [23]. Unfortunately, these differences were not detectable in the sensory analysis. The microbial degradation of pyrazines is still not fully investigated; therefore, to our knowledge, the appearing organisms were not yet described for pyrazine degradation [23]. Since higher concentrations of HMF were measured in samples with additional carbohydrate source, *Z. rouxii* might be responsible for HMF formation [24].

In the present study, we could show that the addition of carbohydrate sources to a moromi fermentation could lead to a temporary change in the microbiota and therefore a shifted aroma profile that was not sensorially detectable. The addition of wheat and buckwheat led to similar results and therefore appeared to be both equally suitable carbohydrate sources. We assume that higher levels of carbohydrates may promote the persistence and flavour formation of *Z. rouxii* in the moromi.

## Figures and Tables

**Figure 1 foods-12-00197-f001:**
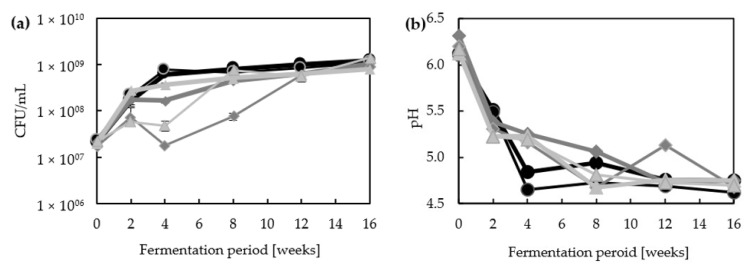
Abundance of microorganisms after four days of incubation on ME agar at 30 °C (**a**) and pH values (**b**) during moromi fermentation. Koji was prepared using lupine seeds with additional roasted lupine seeds (ML1 and ML2, circles, black), wheat (MW1 and MW2, squares, dark grey), or buckwheat (MB1 and MB2, triangles, light grey). The replicates could be distinguished by a grey border (ML2, MW2, MB2). Error bars indicate the standard deviation from quadruplicates.

**Figure 2 foods-12-00197-f002:**
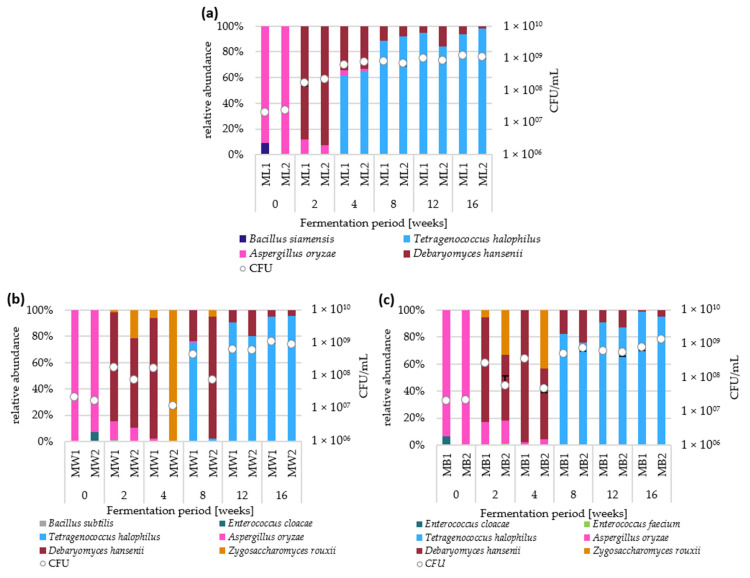
Microbiota dynamics during moromi fermentation, incubated for four days on ME agar at 30 °C via MALDI-TOF-MS. Species and CFU/mL (white dots) in moromi fermentation with (**a**) Lupine seeds as sole substrate, with (**b**) wheat and (**c**) buckwheat. Error bars indicate the standard deviation from quadruplicates.

**Figure 3 foods-12-00197-f003:**
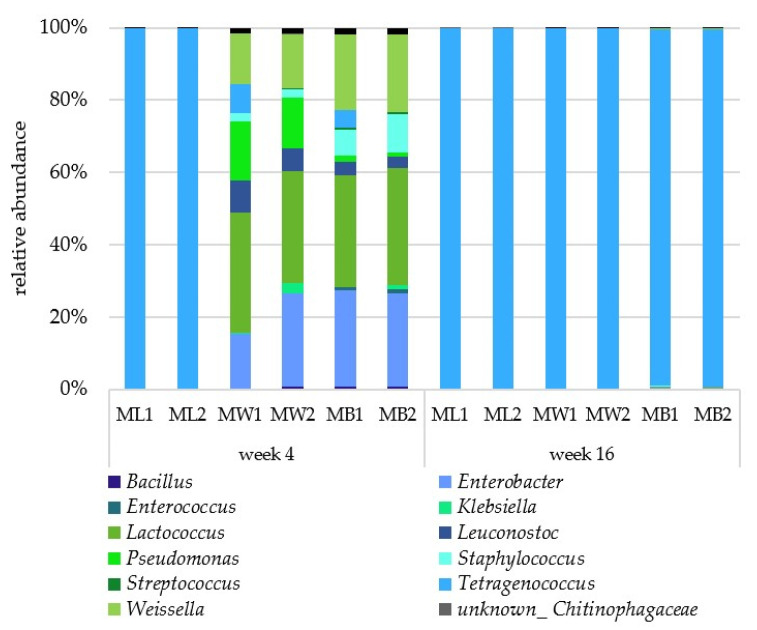
Diversity of bacteria during moromi fermentation determined via 16S rDNA amplicon sequence analysis. Data were evaluated using IMNGS and RHEA [14,15].

**Figure 4 foods-12-00197-f004:**
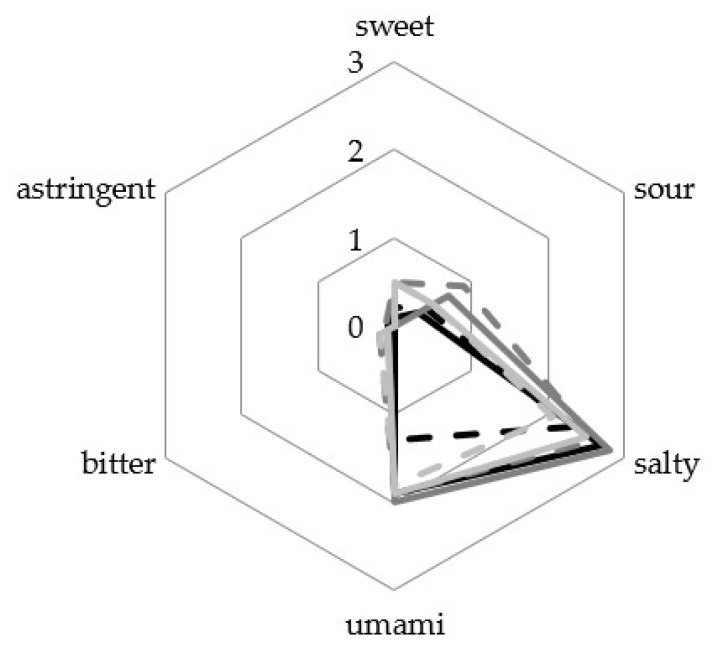
Sensory analysis of the taste of resulting seasoning sauces. Values of moromi with lupine seeds is shown as black lines (ML1) and dashed lines (ML2), with wheat as dark grey lines (MW1) and dashed lines (MW2), and with buckwheat as light grey lines (MB1) and dashed lines (MB2). Data are presented in a 6-point hedonic scale.

**Figure 5 foods-12-00197-f005:**
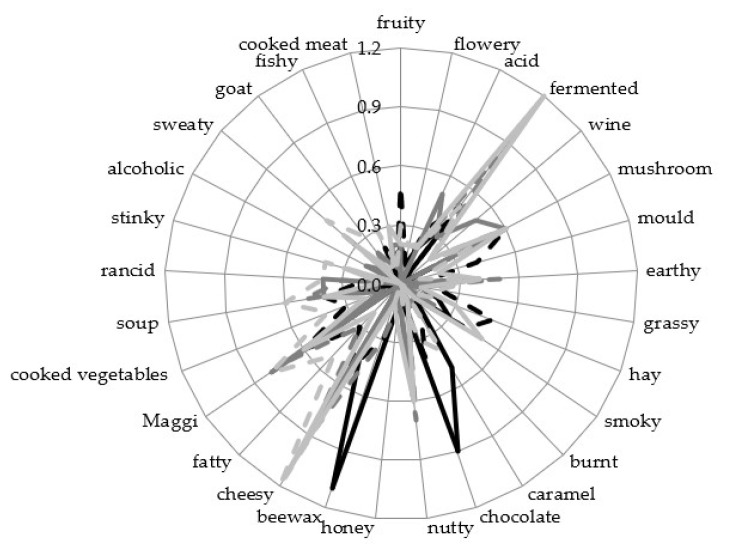
Sensory analysis of the aroma attributes of resulting seasoning sauces. Values of moromi with lupine seeds is shown as black, continuous lines (ML1) and dashed lines (ML2), with wheat as dark grey, continuous lines (MW1) and dashed lines (MW2), and with buckwheat as light grey, continuous lines (MB1) and dashed lines (MB2). Data are presented in a 29-point hedonic scale.

**Table 1 foods-12-00197-t001:** Excerpt of volatile aromatic compounds measured via GC-MS *headspace* analysis after 0, 12, and 16 weeks of fermentation.

	Moromi with Lupine Seeds			Moromi with Wheat			Moromi with Buckwheat		
Compound	0	12	16	0	12	16	0	12	16
2-Methylpropanoic acid		X	X		X	X		X	X
3-Methylbutanoic acid		X	X		>	>		X	X
Acetic acid		X	X		X	>		X	X
Benzene acetic acid		<	X		X	X		X	X
n-Decanoic acid		X	>						
Nonanoic acid	<	X			<			>	
Octanoic acid	X	X	>	X	<	<		>	
2-Methylbutanoic acid ethyl ester					X	X		X	X
2-Methylpropanoic acid ethyl ester					X	X		X	X
2-Phenylethyl acetate					X	X		X	<
3-Methyl-1-butyl acetate					X				<
3-Methylbutanoic acid ethyl ester					>	X		X	X
3-Methylbutyl 2-methylbutanoate					<			>	X
3-Methylbutyl 3-methylbutanoate					<	X		X	X
3-Methylbutyl butanoate					>	<		X	
Ethyl 9-octadecenoate			>		X	X		X	X
Ethyl Acetate					X			X	
Ethyl benzene acetate					X	>		<	X
Ethyl hexadecanoate		X	>		X	X		X	X
Isopentyl 2-methylpropanoate					>	>		<	X
Linoleic acid ethyl ester		X	X		X	X		X	X
Octanoic acid ethyl ester					<			X	
Phenylethyl 3-methyl-butanoate					>	>		<	X
β-Phenylethyl butyrate					X	X		X	X
Dimethyl ether					X	X		X	<
2,3,5-Trimethyl-6-ethylpyrazine	X	X	X	X					
2,3-Dimethyl-5-ethylpyrazine	>	<		>					
2,3-Dimetyhlpyrazine	X	X	X	X		>	X		
2,5-Diethylpyrazine	X	X	<						
2,5-Dimethylpyrazine	X	X	X	X	X	X	X	X	X
2,6-Diethylpyrazine	>	X	X						
2,6-Dimethylpyrazine	X	X	X	X		>			
2-Ethyl-3-methylpyrazine	X	X	X	X	>	>	X		
2-Ethyl-5-methylpyrazine	X	X	X	X	>	>	X		
2-Ethyl-6-methylpyrazine	X	X	X	<	X	X		X	X
2-Methyl-5-((E)-1-propenyl)-pyrazine	>	<	<						
2-Methyl-5-((Z)-1-propenyl)-pyrazine	X	<							
3,5-Diethyl-2-methylpyrazine	X	X	X						
3-Ethyl-2,5-dimethylpyrazine	X	X	X	X					
5H-5-Methyl-6,7-dihydrocyclopentapyrazine	X	X	X	X	>	>	X		<
6,7-Dihydro-2,5-dimethyl-5H-cyclopentapyrazine	X	<	>						
Ethyl pyrazine	X	X	X	X	>		X		
Methylpyrazine	X	X	X	X			X		
Trimethyl pyrazine	X	X	X	X	X	X		X	X
2,3-Dihydrobenzofuran		>	X				X		
5-Isopropyl-3,3-dimethyl-2-methylene-2,3-dihydrofuran		>							
Dihydro-5-((Z)-2-octenyl)-2(3H)-furanone		X	>		X	X	<	>	X

X Compound was detected in both replicates of the fermentation. > Compound was detected in the first replicate. < Compound was detected in the second replicate.

**Table 2 foods-12-00197-t002:** Furanone analysis via HPLC, measured in technical duplicates. The moromi was sampled after four months of incubation with lupine seeds as sole substrate (ML), wheat (MW), and buckwheat (MB).

	HDMF (µg/g)	HMF (µg/g)
ML1	1.47	1.74
ML2	0.92	1.16
MW1	0.76	2.67
MW2	0.60	3.62
MB1	0.83	2.68
MB2	1.87	4.52

## Data Availability

The data related to this study are available on request from the corresponding author.

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
