# Peer review of "Carbohydrate Sources Influence the Microbiota and Flavour Profile of a Lupine-Based Moromi Fermentation"

_foods, 2023, doi:10.3390/foods12010197_

Round 1
Reviewer 1 Report
In introduction you could give more references who could studied on the moromi or soy sauce fermentation
16 S analysis should be explained in detail. Which primers were used in analysis or what are the PCR conditions etc.
Author Response
Thank you for your revision. We tried to improve our manuskript according to your suggestions:
To make it easier for the reader to put this work in context with others, we added relevant studies as references to the manuscript.
We added the used Primers and another reference to clarify the method. Since we already used the same procedure in a previous publication, we tried to simplify this section.
Reviewer 2 Report
About the abstract. It would be interesting to show if the starter culture was an independent treatment from the treatments of the carbohydrate sources, o if the starter culture was added to both carbohydrate sources. This comment also has sense, because there is no information in the abstract about the effect of the starter culture on the taste.
About the introduction. It is necessary to add information about the acceptance quality characteristics of fermented seasoning sauces, even about the lupine-based seasoning sauce. Either perception from the local consumers, or from other studies made about these products.
Author Response
Thank you for your revision. We tried to improve our manuskript according to your suggestions:
The starter cultures were added in all batches independent from the carbohydrate source. Effects on the moromi are only visible by comparison with the literature. There is no published data about the taste of lupine sauce outside of this study; Therefore, the effect of the starter cultures on the taste is not yet determined. The abstract was misleading in this point so we edited this phrase.
Since there is a variety of different seasoning sauces and preferences, there is no optimum for the aroma (except for the overall goal of umami taste). As a result, we don’t want to dictate which attributes are necessary in the product. We tried to explain the influence of different organisms on the flavor to give hints on how it can be modified e.g. by using starter cultures.
Reviewer 3 Report
Fermented seasoning sauces are very popular all over the world. Lupine-based seasoning sauce is produced similar to soy sauces and therefore generates a comparable microbiota and aroma profile, but the usage of a yeast starter culture resulted in a sparsely divers microbiota and led to a pH below 5. So in this study, the authors monitored the influence of a yeast starter culture and the addition of a carbohydrate source on the microbiome and aroma profile of the resulting sauce. This study is very interesting. However, the importance of the target is unclearly throughout the whole manuscript. The major comments are listed below:
1. It is popular to use soy and fish and so on to ferment seasoning sauces. The authors always to introduces how to produce seasoning sauces and the microbiota participate in the fermention process, but did not introduce the specific contents and why will do this study in the Introduction? The introduction needs to be re-organized.
2. In Materials and Methods, “Each fermentation was prepared in duplicates……”Why prepared in duplicates not triplicates? Please explain.
3. In 2.2, in the second paragraph of this section, all “ml” must change to “mL”.
4. There is no error-bar in figure 1, why?
5. In section 3.1, with addition carbohydrate source, cell counts were lower in batches while reached cell counts of similar compared to the control group after 16 weeks, is it a normal phenomenon? And have any reference to prove? Please add some references and explain.
6. Sensory analysis is very interesting, it can description the flavor. But from the final result, the lupine sauces with additional carbohydrate sources did not show significant differences. How to make sense of this sentence?
7. What is the conclusion of this study and what is the prospect of this study? The authors did not mention at all. Please supply.
Author Response
Thank you for your constructive review. We tried to clarify the importance of this study with adjustments in the introduction and discussion and tried to edit your recommended revision:
- We included more information about why we used lupine seeds and the necessity for this research. Unfortunately, we do not have the information about the exact composition of the used lupine seeds.
-
We know that triplicates are usually used to validate results. However, the sampling of duplicates where time and money consuming and we were not able to handle triplicates in this experiment. We have taken our cue from other works, where triplicates were also frequently omitted (Tanaka et al. 2012, Yan et al. 2013, Song et al. 2015).
-
This was corrected as recommended.
-
We added the error-bars in this figure.
-
We couldn’t fully explain this phenomenon, but it appears to be connected to the presence of Z. rouxii. The cell counts of Z. rouxii were similar but in some cases, cell counts of D. hansenii were decreased (which shows in the CFU sum). We tried to explain this in the second paragraph of the discussion and to our knowledge there is no study of cocultivation of these yeasts. A decrease in cell counts after the first week was a common phenomenon we could observe in batches without starter cultures (Lülf et al. 2021).
-
This is the first study that shows sensory analysis of a lupine-based seasoning sauce. Even without significant differences, we wanted to show these data since they are still a novelty in this field. Moreover, we can’t exclude other flavor attributes that were not tested to show differences between the sauces.
-
We added a final conclusion with our prediction to the discussion, hoping to illustrate the importance of this study.